# The sources of atmospheric black carbon at a European gateway to the Arctic

P. Winiger[1], A. Andersson[1], S. Eckhardt[2], A. Stohl[2] & Ö. Gustafsson[1]

Black carbon (BC) aerosols from incomplete combustion of biomass and fossil fuel contribute to Arctic climate warming. Models—seeking to advise mitigation policy—are challenged in reproducing observations of seasonally varying BC concentrations in the Arctic air. Here we compare year-round observations of BC and its $\delta^{13}C/\Delta^{14}C$-diagnosed sources in Arctic Scandinavia, with tailored simulations from an atmospheric transport model. The model predictions for this European gateway to the Arctic are greatly improved when the emission inventory of anthropogenic sources is amended by satellite-derived estimates of BC emissions from fires. Both BC concentrations ($R^2 = 0.89$, $P < 0.05$) and source contributions ($R^2 = 0.77$, $P < 0.05$) are accurately mimicked and linked to predominantly European emissions. This improved model skill allows for more accurate assessment of sources and effects of BC in the Arctic, and a more credible scientific underpinning of policy efforts aimed at efficiently reducing BC emissions reaching the European Arctic.

[1] Department of Environmental Science and Analytical Chemistry, and the Bolin Centre for Climate Research, Stockholm University, Svante Arrhenius väg 8, Stockholm 10691, Sweden. [2] Department of Atmospheric and Climate Research, Norwegian Institute for Air Research, NILU, Instituttveien 18, Kjeller 2027, Norway. Correspondence and requests for materials should be addressed to Ö.G. (email: orjan.gustafsson@aces.su.se).

Black carbon (BC) is the most important type of light-absorbing aerosol, and contributes substantially to a positive radiative forcing on the global climate[1,2]. Even though the atmospheric concentrations of BC in remote areas—such as the Arctic—are low in general, their effects on regional climate may be substantial[3–5]. Current atmospheric chemistry-transport and climate models both underestimate the loadings of BC and fail to reproduce much of the observed seasonality of Arctic BC concentrations observed at ground-based stations[6,7]. The underlying reasons for the offsets between measurements and model predictions of BC in the Arctic are currently unclear. Possible explanations include uncertainties in Arctic meteorology, aerosol lifetimes and emissions[6]. A recognized complication for climate and chemical-transport modelling of BC is the large uncertainties associated with technology-based emission inventories (EIs)[8–10]. This is illustrated by the fact that the relative contributions of biomass burning versus fossil fuel combustion predicted by EI models do not agree with [14]C-based diagnostic source apportionment of BC in the actual atmosphere, at least not for South Asia[11,12] and East Asia[10,13]. Hence, a major challenge in accurately assessing BC climate effects may stem from uncertainties in BC EIs.

Bottom-up EI estimates are calculated as the product of the activity (that is, amount of burnt fuel) and the emission factor (that is, amount of BC released per amount of burned fuel). EIs are relatively reliable for the most important greenhouse gas, $CO_2$. However, for products of incomplete combustion, such as BC, the uncertainties are larger for both the activity factor and especially for the highly variable emission factors—particularly for solid biofuel combustion and open combustion processes[8]. The most important emission regions for the Arctic are believed to be the latitudes between 30 and 60°N (refs 7,14). However, there are large uncertainties regarding the BC EI at higher latitudes (north of 60°N) such as from temporally varying tundra and taiga wildfire emissions, and the gas flaring emissions of the petroleum industry[7,15,16]. Combined with other complicating factors—such as in transport simulations, BC aerosol aging, removal efficiency and consequently lifetime[1,6]—several recent assessments call for observationally based source evaluation to refine BC EIs[1,3,5,16–19].

Top-down, dual-carbon isotope-based—that is, [14]C/[12]C (radiocarbon) and [13]C/[12]C (stable carbon)—characterization of elemental carbon (EC) aerosols (the mass-based analogue of optically-defined BC) has in recent years proven its value for quantitatively constraining contributions from different BC emissions sources in Asia[10,11,13,20] and the Arctic[21,22].

Here, we present a year-round [14]C-EC study of the Arctic. The measurement site was located in Abisko, in the Swedish Arctic, which is a gateway for Eurasian emissions to the high Arctic[23]. Samples were collected using high-volume aerosol samplers for two size fractions, $PM_{2.5}$ (particulate matter with an aerodynamic equivalent diameter smaller than 2.5 μm) and TSP (total suspended particles). Both stable- and radiocarbon isotope analyses were focused on the $PM_{2.5}$ EC fraction, and the two-dimensional isotopic signature was subjected to statistical modelling using Markov-Chain Monte Carlo (MCMC) simulations[13,24] to account for the uncertainties in the source signatures. These observational data were directly compared with a tailored Lagrangian particle dispersion model (FLEXPART, FLEXible PARTicle dispersion model), coupled to the recently developed ECLIPSE (Evaluating the Climate and Air Quality Impacts of Short-Lived Pollutants) EI[16]. Temporally varying fire emissions were included in the model using daily satellite data implemented in the Global Fire Emissions Database (GFED)[25] at monthly resolution. This allows for a diagnostic comparison, between the measured (top-down) and simulated (bottom-up)

BC contributions from different source types, that is, different fossil fuels versus contemporary fuels (including biofuel and open biomass burning). We find that the dominating source of observed BC is of European origin and consists to 45% of biomass burning sources, followed by liquid fossil fuel combustion (35%) and coal combustion (20%). Further, the comparison between model and observation is significantly improved by inclusion of open biomass burning in the model. This proof of concept has the potential to become a valuable tool for studying BC in the Arctic, to improve future climate modelling scenarios.

## Results

**Receptor site and meteorological setting.** The Abisko research station (ANS; operated by the Swedish Polar Research Secretariat), is located 200 km north of the Arctic Circle (Fig. 1) and has been a hub for climate and ecological research for well

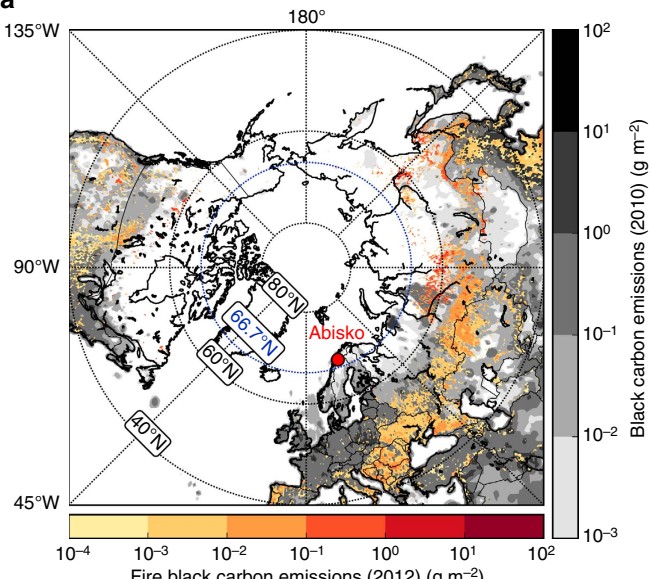

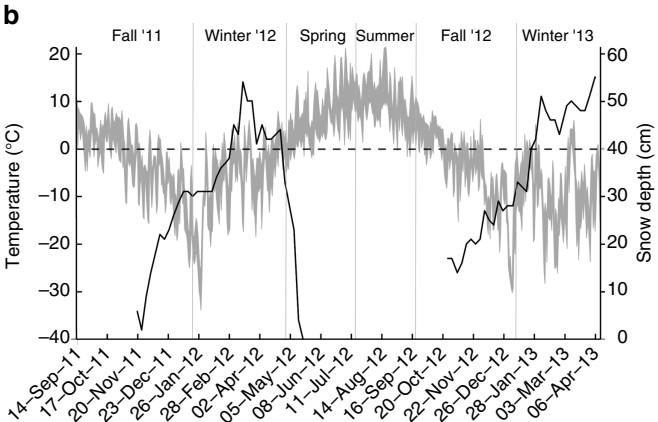

**Figure 1 | Location of Abisko and its surrounding BC emissions and meteorological conditions during the sampling campaign.** (**a**) Abisko (red dot), surrounded by anthropogenic BC emissions (baseline scenario 2010) from the ECLIPSEv5 EI (grey log scale) and fire BC emissions of open fires by GFED4.1s (red log scale) for the year 2012. (**b**) Temperature (grey shaded area) and snow depth (black line), measured at Abisko Scientific Station, 10 km west of the sampling station (100 km north of the Arctic Circle). Zero degree Celsius is marked with a dashed line.

over 100 years[26]. Stordalen mire—the study site—lies 10 km east of the ANS at the southern shore of Lake Torneträsk in the Scandinavian mountains (Supplementary Fig. 1) and belongs climatically to the sub-Arctic, with an annual mean temperature close to 0 °C and relatively little precipitation[26]. The study took place from 9 September 2011 to 27 March 2013, where the main study year (2012) was slightly colder than a 20 year-average (0.2 °C during 1993–2013), with a mean of –0.9 °C, and with an accumulated 367 mm in precipitation, similar to the 20 year mean (1993–2013) of 354 mm (Fig. 1b).

**Carbon aerosol concentrations.** Concentrations of EC are typically elevated in the winter, while organic carbon (OC) is elevated in the summer. OC may come from both primary and secondary sources, and these, in turn, can be from both combustion and non-combustion sources. High OC/EC ratios are therefore a rough indicator for the contribution of biomass burning and secondary aerosol formation from biogenic volatile organic compounds, which is a well-known phenomenon for the boreal belt[27] (Supplementary Fig. 2). Winter pollution events are often referred to as Arctic Haze and are connected to a contracted boundary layer, pollution influx from lower latitudes into the polar dome, and inefficient removal processes[23]. Trends show that BC concentrations have generally been decreasing in the last couple of decades with rates of 2% per year in the European Arctic[28,29].

Measurements of both TSP and fine fraction aerosol (PM$_{2.5}$) were in good agreement with each other ($R^2 = 0.75$ for EC and $R^2 = 0.78$ for OC; both $P$ values are $< 0.05$; all coefficients of determination ($R^2$) used in this work are from linear regressions) and followed the same temporal trends during the campaign (Supplementary Fig. 3). The EC PM$_{2.5}$-to-TSP ratio (that is, EC fine fraction) was $76 \pm 23\%$ for the whole period ($77 \pm 24\%$ for both winters). In the following discussion, the focus is on PM$_{2.5}$, primarily because the source-diagnostic isotope data are available for the whole study period and secondarily because the model predictions are based on the fine fraction. The observed PM$_{2.5}$ EC concentrations in Abisko showed an annual average for the year 2012 of 27 ng C m$^{-3}$ with large seasonal variability, including maxima in the two observed winters of 130–160 ng C m$^{-3}$ (Fig. 2a).

Yearly and quarterly averaged BC particle concentrations were in a similar range to nearby stations, although sometimes different types of measurements have been applied (for a general clarification of nomenclature see Petzold *et al.*[30]). To ease comparison, the seasons were defined analogously to earlier studies (Supplementary Table 1). At the nearby Finnish Kevo station (~350 km ENE of Abisko) with a 47-year record of BC measurements, average TSP EC concentration of 67 ng C m$^{-3}$ for 2010 and ~100 ng m$^{-3}$ for the 2001–2010 period were found[28]. Archived Kevo samples from the most recent 30 years were analysed with the same method used in the present study (thermal-optical transmission (TOT), NIOSH 5040) while older ones (from before 1979) were analysed with an earlier version of the recently used protocol. The EC concentrations from Abisko and Kevo compared quite well and differ by only 40–60%. Compared with the slightly higher Kevo values, Abisko appears to be little influenced by the ore smelting and mining industry on the Kola Peninsula[31]. This is further consistent with the FLEXPART-derived geographical source information suggesting near absence of footprints from that region (Supplementary Fig. 4).

The Finnish GAW (Global Atmosphere Watch) station Pallastunturi (~200 km E of Abisko) reported annual mean equivalent BC (EBC) of $64 \pm 103$ ng m$^{-3}$ during 2007 and 2008

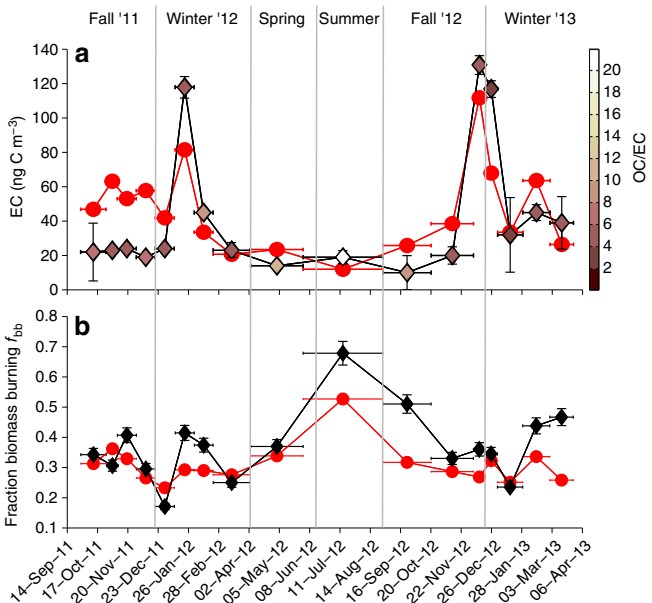

**Figure 2 | Comparison of BC levels and sources between observations and model predictions.** Horizontal bars indicate duration of sampling. Vertical bars indicate s.d. of the observational data (**a**), Bottom-up model predictions from FLEXPART (red line and round symbols). Elemental carbon concentration for PM$_{2.5}$ EC concentrations from the top-down observations (black line and diamond symbols). The colour bar represents the OC/EC fraction for each PM$_{2.5}$ sample (diamonds). (**b**) BC source apportionment expressed as fraction of biomass burning ($f_{bb} = 1$—fraction fossil) for top-down measured PM$_{2.5}$ (black line and diamond symbols) and bottom-up BC simulated with FLEXPART (red line and round symbols). The $f_{bb}$ uncertainties (s.d.) shown for the PM$_{2.5}$ based $f_{bb}$ are the results of the MCMC calculations (Supplementary Table 5).

(size cutoff PM$_{10}$ or bigger)[32]. Annual means for other Arctic stations were also similar to Abisko. Alert (Nunavut, Canada) records show ~50 ng m$^{-3}$ BC (aethalometer data adjusted to EC) for the period of 1997–2007 (ref. 33) and Zeppelin Observatory (Svalbard, Norway) records show 39 ng m$^{-3}$ BC (aethalometer data) for the period of 1998–2007 (ref. 34). The observed concentrations in Abisko can thus be considered as pristine Arctic background values.

**Carbon-isotopes-based EC source apportionment.** Characterization of the dual-carbon isotope signature ($\delta^{13}$C and $\Delta^{14}$C) of carbonaceous aerosols provide direct quantification of the relative contribution from major emission source classes. The $^{14}$C-based fossil versus contemporary constraints (equation (1)) show that on an annual average (2012) the relative contribution of biomass burning to EC in the fine fraction was $42 \pm 15\%$. However, a large seasonal variability in the fraction of biomass burning contribution was observed throughout the year, with high values in the summer period (up to 68%) and lower values in the winter (down to 17%). Stable isotope values of EC show no clear seasonality, fluctuating also within the seasons. The most depleted (–27.9‰) and most enriched (–24.1‰) $\delta^{13}$C values were observed consecutively in the winter of 2012. Nonetheless, $\delta^{13}$C values indicate slight shifts in sources throughout the year (Fig. 3). Expected combustion sources are predominantly liquid fossil fuels, biomass and coal (see Supplementary Table 2 for isotopic signatures of fuels, that is, source end members). In general, the majority of EC at Abisko ($76 \pm 23\%$) was found in the fine PM$_{2.5}$ fraction throughout the campaign. It is therefore not surprising that the

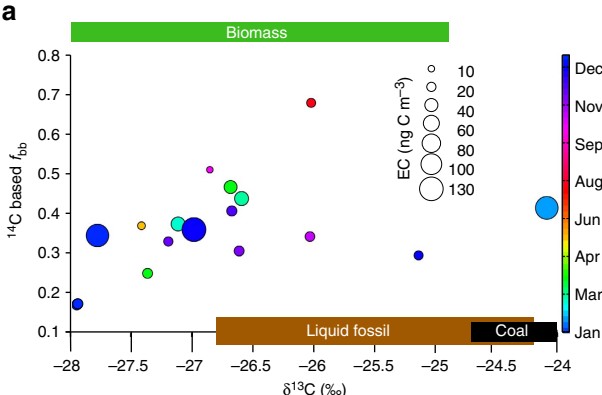

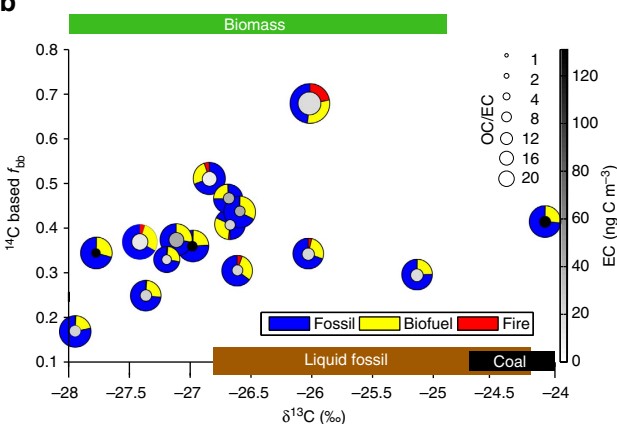

**Figure 3 | Multi-dimensional source apportionment.** The expected two-dimensional $\delta^{13}C$ and $\Delta^{14}C$ endmember ranges for biomass burning emissions, liquid fuel combustion and coal combustion are shown as green, brown and black bars, respectively. (**a**) Seasonal variations (colour bar), respresented by the coloured shading within the circles. The circles' area represent EC concentrations in $ng\,C\,m^{-3}$ (see scale with black circles on upper-right side). (**b**) The circles (same as in **a**) in grey scale show the respective EC concentrations, their area represent OC/EC ratios from 2 to 22 (see scale with black circles on upper-right side). The coloured rings represent fractions of fossil (blue), biofuel (yellow) and open fires (red) based on FLEXPART (see Supplementary Table 9).

$\delta^{13}C$ and $\Delta^{14}C$ signatures in the total and fine fractions are rather similar (Supplementary Table 3).

**Prediction vs observation of concentrations.** The year-round Abisko campaign provides an opportunity to directly compare predictions of both BC total concentrations and contributing emission sources from a transport model (FLEXPART, driven with ECLIPSE emissions and monthly open fire emissions from GFED) with observations of concentrations and isotope-based source diagnostics. Encouragingly, the model simulations captured the overall observed ($PM_{2.5}$ EC) concentrations and seasonality for EC in the Swedish Arctic well ($R^2 = 0.61$; $P < 0.05$; Fig. 2). Predicted concentrations are only biased high (factor of ~2) for the beginning of the observation period (fall 2011). However, the source estimates for this time-period are in good agreement (see more below), and the geographical sources contributing to this period are similar to other periods. We can therefore only speculate that this offset relates to uncertainties in the EIs, for example, its coarse monthly resolution. For the full year of 2012, the coefficient of determination between measured and simulated concentrations increased to 0.89 ($P < 0.05$). The

two highest simulated BC (and observed $PM_{2.5}$ EC) concentration values occurred in the two winters, when the air predominantly was transported from Eastern Europe and across Scandinavia, thus accumulating BC emissions from the European continent. On a quarterly (seasonal) basis, the modelled and observed BC concentrations are almost identical, with the FLEXPART model (Supplementary Table 4), thus showing good model skill in capturing the observed large BC seasonality, with some over-predictions in the two fall periods. The winter concentrations were well predicted in both years, a period when the observed contribution of fossil fuel combustion rose to local maxima.

**Prediction vs observation of sources.** The quantitative isotope-based EC source apportionment provides credible verification of improved skills in the modelling of Arctic BC, as not only the loadings but also the relative contribution from different sources must be accurate. There was also good model–observation agreement, including seasonal variability, in the detailed BC source apportionment. Contributions from fraction biomass burning ($f_{bb}$) emissions were only slightly under-predicted ($R^2 = 0.57$; $P < 0.05$) with a model-predicted average $f_{bb}$ of $34 \pm 6\%$, compared with an observation-constrained average $f_{bb}$ of $40 \pm 12\%$.

The seasonality observed in the EC sources was also captured by the model (Fig. 2b). The observation-based ($PM_{2.5}$ EC) $f_{bb}$ showed a large amplitude and ranged from wintertime low of $35 \pm 10\%$ (January–March 12) and $38 \pm 9\%$ (January–March 13) to spring–summer values of $58 \pm 15\%$ (April–June and July–August 12), with an average for the year 2012 of $f_{bb} = 42 \pm 15\%$. The model predictions for the full year 2012 yielded a $f_{bb}$ of $35 \pm 9\%$, which thus was in good agreement with the observed annual average $f_{bb}$ ($R^2 = 0.77$; $P < 0.05$). A main reason for the accurate (seasonality) prediction of the current model is likely the inclusion of open fire contributions. Without considering vegetation fires and agricultural waste burning, the coefficient of determination ($R^2$) would only be 0.26 ($P$ value = 0.05). This suggests that the high variability in open burning and forest fires are crucial in capturing and reproducing the observations.

**Geographical sources.** The general agreement between observations and modelling suggests that the FLEXPART model, when driven by updated estimates of both anthropogenic and wildfire emissions, can be used with confidence to examine the major geographical source regions affecting the European Arctic. The present FLEXPART-ECLIPSE BC footprint emissions suggest Europe as major geographical source region for the Abisko site (Supplementary Fig. 5), while other studies suggest that Asian BC burdens are generally dominating in the Arctic[19], especially at higher altitudes[35]. The European source can be divided into three main zones: the eastern European region (for example, Poland, European Russia and the Baltic states), the western European region (for example, Benelux and UK) and the northern European region (for example, Sweden, Finland, Norway and Denmark). Although emissions from the towns of Kiruna and Narvik (~70 km away) may contribute some to the BC loadings in Abisko, the majority of BC is clearly from long-range transport (~1,000–2,000 km; Supplementary Fig. 6), as also suggested by the fact that Abisko has among the lowest BC loadings of all pan-Arctic observatories.

The highest BC loadings observed during the campaign were re-occurring, fossil-rich, wintertime emissions from the eastern European region (Fig. 2), in accordance with observations in Pallastunturi during winter 2012 (ref. 36). In addition to geographic variability in air mass transport pathways, seasonality is affected by occurrences of vegetation fires. The

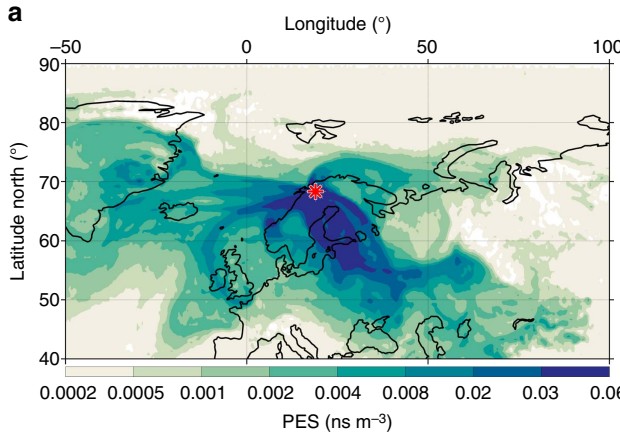

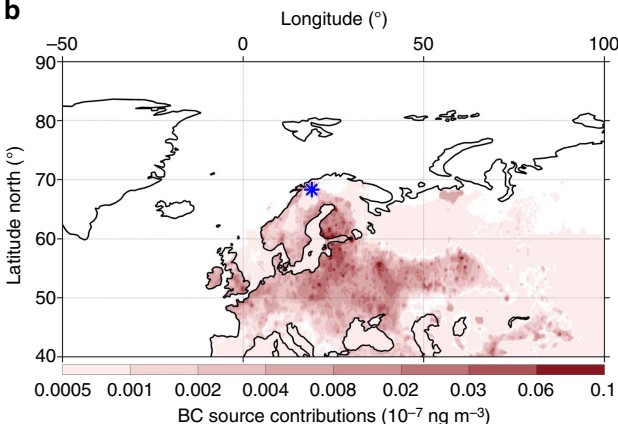

**Figure 4 | FLEXPART footprint and source contribution.** Model output for the most $\delta^{13}$C-enriched sample (2012-01-11 to 2012-02-01). The applied scale is the logarithm to the base of 2 (log2). (**a**) PES for the BC aerosol arriving at Abisko (red star). (**b**) Geographical distribution of the anthropogenic BC source contribution to the simulated mixing ratio at Abisko (blue star). PES, potential footprint emission sensitivity.

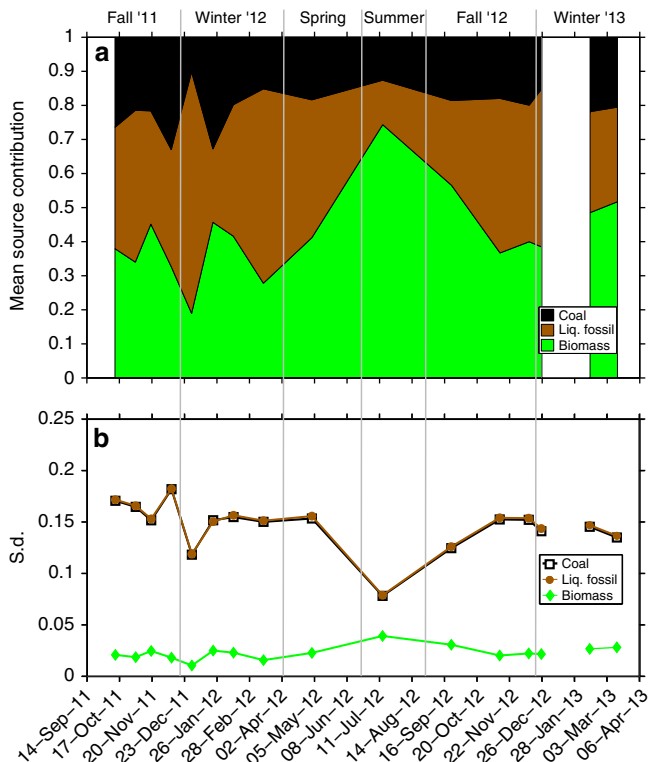

**Figure 5 | Dual-isotope-constrained source contributions from the Bayesian MCMC model.** (**a**) MCMC based source contribution of fossil coal combustion, liquid fossil fuel combustion and biomass burning. Black (upper) area shows the coal mean; brown (middle) shows the liquid fossil fuel mean; and green (lowest) area shows the biomass mean. (**b**) The corresponding s.d. to the respective values in **a** for coal (white squares), liquid fossil fuel (brown circles) and biomass (green diamonds). The white gap is due to one sample missing $\delta^{13}$C data.

vegetation fires are more prominent in the drier summer season, and predominantly occur in the eastern European region (Fig. 1 a).

**Bayesian MCMC model.** This statistical approach accounts for the variability of the $\delta^{13}$C and $\Delta^{14}$C endmembers and allows a statistical apportionment between three source classes: biomass burning, coal combustion and liquid fossil fuel combustion (Supplementary Table 5)[13]. The results of the MCMC model are the posterior probability density functions for the relative contribution of the three sources. The evolution of the fraction of biomass burning EC ($f_{bb}$) is similar to that obtained from the previously discussed radiocarbon data. The additional information derived by MCMC of the $\delta^{13}$C and $\Delta^{14}$C data, is the further split of the fossil fraction into coal ($f_{coal}$) and liquid fossil fuel ($f_{liq.fossil}$). Since these two have almost equal uncertainties in their isotopic signature (Supplementary Table 2) their s.d. become very similar, whereas the s.d. for the biomass fraction is much smaller due to a lower uncertainty. The most enriched $\delta^{13}$C signature ($-24.1$‰, 11 January 2012) was observed during one of the high loading events originating from the Eastern European region (Fig. 4), with a dual-isotope-based estimation of the coal contribution at 33% (60% of the fossil contribution, Fig. 5). For the other samples, the liquid fossil

fraction—with high seasonal amplitude ($f_{liq.fossil}$: 0.13–0.71)—is larger or equal to the coal fraction. Coal combustion is with an average of 20% still a considerable EC source and shows much smaller seasonality ($f_{coal}$: 0.10–0.31) than the other two EC fractions. Seasonal averages depict a clear hierarchy with biomass burning dominating EC followed by liquid fossil fuel and coal combustion. However, except for summer the two (anthropogenic) fossil fuel sources together are the largest contributing source, with roughly 55% over the year 2012 or the whole 18-months sampling period (Supplementary Table 6).

**Discussion**

The quantitative isotope-based EC source apportionment provides support of improved skills in the modelling of Arctic BC, as both the concentration loadings and the relative contribution from different sources are in agreement. Earlier transport model simulations, only performed for total concentrations, have been waning when compared with observations in the Arctic. It is likely that systematic biases and uncertainties in BC EIs, possibly enhanced by challenges in modelling of the transport in the Arctic troposphere and in the scavenging of aerosols, have caused the earlier model-observation mismatch of BC in the Arctic. The present isotope-constrained source apportionment study now demonstrates the ability of the FLEXPART model, with improved description of BC emissions, to reproduce both the absolute concentrations and their seasonal amplitude, as well as assigning the contribution of different source classes to the simulated BC in

agreement with the observed source diagnostics. Only concentrations in the fall period were overestimated by a factor of two, but the strong seasonality was well captured by the simulations. The FLEXPART-ECLIPSE-GFED model was also used to compare model predictions with our previous carbon-isotope-based source apportionment study for BC at the Zeppelin observatory on Svalbard[21]. For that limited study (January–March 2009), larger offsets were observed between model and observations (Supplementary Fig. 7 and Supplementary Note 1). A major reason for this is likely a bias in the collected samples, as that study was focusing specifically on high-pollution events, due to low filter loadings available from short sampling durations (24 h). This comparison thus remains inconclusive, but perhaps suggests the influence of temporally varying local sources at Zeppelin, and called for longer and continuous observational periods such as in the current 18-months study. The improved ability by the transport model that was used in the current long-term study with monthly resolution of BC emissions and by including monthly open fire emission, suggest that the earlier reported mismatches between simulations and observations can be clearly related to and partially rectified by improved emission information and not only by proper tuning and parameterization of physical processes such as removal.

This isotope-observation-enabled validation of improved model skill for Arctic BC opens up for both more accurate assessment of sources and effects of BC in the Arctic, and a more credible scientific underpinning of efforts aiming at efficiently reducing BC emissions reaching the European sector and possibly the greater Arctic.

## Methods

**The Swedish Arctic sampling campaign.** The sampling site was located at (68.36N, 19.05E, 359 m above sea level) a 25 ha nature reserve 10 km east of Abisko village (<100 year-round inhabitants). Samples were collected continuously from late September 2011 to March 2013 with filter-changing intervals of 12–28 days, depending on the season and weather conditions, ensuring sufficient EC mass to allow for microscale $^{14}$C analysis. Aerosols were collected on precombusted quartz fibre filters (Millipore) using parallel high-volume sampling instruments with $PM_{2.5}$ (model DH77, Digitel AG) and TSP inlets (custom built at Stockholm University).

**Elemental carbon and organic carbon analysis.** Carbonaceous aerosol concentrations (EC and OC) were measured at Stockholm University with a standard TOT analyser (Sunset Laboratory Inc.) using the National Institute for Occupational Safety and Health (NIOSH) 5040 method[37]. Potential effects of charring with this method, where parts of the OC could end up as pyrogenic carbon in the EC fraction, have been evaluated in earlier work by sensitivity analysis reaching the conclusion that fraction biomass burning could even in extreme cases be overestimated only by a maximum of 7% (ref. 21). During the whole study period 30 samples (and 17 blanks) for each $PM_{2.5}$ and TSP were collected (Supplementary Table 7). Detection limits for OC aerosols in this study were estimated from field blanks (5% of $PM_{2.5}$ and 2% of TSP mean filter load). There was no EC detected in any of the blanks. The average relative s.d. for triplicate analysis were 3 and 4% for $PM_{2.5}$ and TSP OC, and 8% and 3% for $PM_{2.5}$ and TSP EC, respectively.

**Carbon isotope analysis.** A key advantage of isotope-based EC source apportionment is its representation of total EC and is hence independent of chemical tracers. The latter tends to non-conservative transport behaviour for long-range transport, such as the Arctic, and are not parts of EC isolates[38]. In contrast, the $\delta^{13}$C and $\Delta^{14}$C isotope signatures in EC are thus intensive properties, with the same fate, such as atmospheric lifetime, as the total EC. Hence, these isotope-based source apportionment techniques are particularly well suited for regions like the Arctic[21,22].

Radiocarbon results are often presented on the $\Delta^{14}$C scale, which includes a normalization using a standard value for the $\delta^{13}$C signature. This has the benefit that $\Delta^{14}$C is corrected for atmospheric processing (and its stable isotope fractionation), which in the specific case of recalcitrant EC is negligible. Isotopic values of $^{14}$C/$^{12}$C and $^{13}$C/$^{12}$C are reported as $\Delta^{14}$C and $\delta^{13}$C, respectively, on a per mil scale[39–41]. The relative contributions to atmospheric EC from biomass burning ($f_{bb}$; including biofuel and open burning fires) and fossil combustion

($f_{fossil} = 1 - f_{bb}$) sources were calculated with an isotopic mass-balance equation[11]:

$$\Delta^{14}C = \Delta^{14}C_{bb} f_{bb} + \Delta^{14}C_{fossil}(1 - f_{bb}) \qquad (1)$$

Here, $\Delta^{14}$C represents the radiocarbon signature in the ambient samples. By definition, $\Delta^{14}C_{fossil}$ is $-1,000‰$, since fossil carbon is completely depleted in radiocarbon. Endmembers for contemporary radiocarbon $\Delta^{14}C_{bb}$ depend on type and age of the studied biomass. In the (Swedish) Arctic the most common source of biomass fuel is wood, for which an endmember between $+189$ and $+264‰$ is suggested[21,40,42–44]. This range is narrowed down by Monte Carlo simulations to an endmember of $+225 \pm 25‰$, translating to a variability of $<5\%$ in the resulting calculated fraction of biomass burning EC using MCMC techniques, detailed below[21,24].

Seasonal and yearly means of the fraction of biomass burning EC were calculated as

$$\overline{f_{bb}} = \frac{\sum_{i=1}^{n} f_{bb}(i) \cdot EC(i) \cdot V(i)}{\sum_{i=1}^{n} EC(i) \cdot V(i)} \qquad (2)$$

Where $f_{bb}$ is the fraction of biomass EC, EC the elemental carbon concentration, $V$ the volume collected with the respective sample and $i$ is the sample index.

Seasonal and yearly means for FLEXPART were calculated on the basis of

$$\overline{f_{bb}} = \frac{\sum_{1}^{n} f_{bb}(i) \cdot t(i)}{\sum_{1}^{n} t(i)} \qquad (3)$$

Where $f_{bb}$ is the model-based fraction biomass, and $t$ is the sampling time for the respective sample $n$ is the number of available samples.

To determine the carbon isotopic fingerprints of EC, $PM_{2.5}$ samples were pooled into 17 composites/samples. Higher temporal resolution was chosen during Arctic Haze seasons (winter/spring), and lower resolution was chosen for the summer months, when EC concentrations are lower (see Supplementary Table 3 for sampling start dates of composites and duration of sampling). The EC fractions were cryogenically trapped for further off-line isotopic analysis after regular Sunset-TOT conversion to $CO_2$, as described in previous work[10,12,13,21]. The isolated and trapped $CO_2$ was then analysed for its natural $^{14}$C abundance and $^{13}$C/$^{12}$C ratio using accelerator mass spectrometry (AMS) at the US-NSF NOSAMS Facility (Woods Hole, MA, USA)[40,45,46].

**Bottom-up emission inventory and transport modelling.** For the bottom-up estimates of the BC concentrations at Abisko the atmospheric dispersion model FLEXPART was used[47,48]. FLEXPART version 9.2 was run in backward mode from the station location and for the exact same time-periods over which the measurements were taken. A logarithmic size distribution with mean particulate diameter of 250 nm was used, with a variation of sigma 1.25 (logarithmic s.d.). Simulations extended over 20 days back in time, which is sufficient to include most emissions injected into an air mass arriving at the station, given a typical BC lifetime of the order of a week. The simulations used meteorological analysis data from the European Centre for Medium-Range Weather Forecasts (ECMWF) at a resolution of $1° \times 1°$ latitude/longitude. FLEXPART accounts for dry deposition and wet scavenging, distinguishing between below-cloud and in-cloud scavenging. For anthropogenic BC emission information, FLEXPART was coupled to the ECLIPSE version 5 EI (Baseline scenario for the year 2010)[16], which is based on the the GAINS model (Greenhouse gas—Air pollution Interactions and Synergies)[49]. The emissions were available at a 0.5° spatial resolution and a yearly resolution for various source types and, in addition, contain an explicit split between biofuel (modern; for example, wood burning) and fossil fuel emissions (Supplementary Table 8). Monthly emissions were derived by splitting the annual emissions into twelve components, based on the respective month's duration. Like many other EI, GAINS does not contain uncertainty estimates related to the individual emission types. Other bottom-up EIs report uncertainties in the range of $+125\%$ for BC emissions[9]. For non-agricultural open biomass burning BC emissions (for example, vegetation fires), which are not accounted for by ECLIPSE, emissions estimates based on the GFED were used[25]. Since GFED also includes fires from open agricultural waste burning, these emissions were not taken from the ECLIPSE estimates.

**Open-fire estimate using the GFED fire emission inventory.** To estimate the biomass burning contribution from wild fires and agricultural waste burning, the Global Fire Emissions Database version GFED4.1s, with monthly temporal resolution and a spatial resolution of 0.5° (refs 25,50). This version of the fire EI, based on Collection 5.1 MODIS (Moderate Resolution Imaging Spectroradiometer) burned area product[51], includes small fires[52], observed by active fire detections on board the MODIS Aqua and Terra satellites. The data for conversion to high temporal resolution[53], and a set of emission factors to convert dry matter to BC are provided by GFED as well[54].

**Bayesian statistics.** The dual-carbon isotope data was used in combination with a MCMC technique to further constrain between the three source classes: biomass

($f_{bb}$), liquid fossil ($f_{liq.fossil}$) and coal ($f_{coal}$)[13].

$$\begin{pmatrix} \Delta^{14}C \\ \delta^{13}C \\ 1 \end{pmatrix} = \begin{pmatrix} \Delta^{14}C_{bb} & \delta^{14}C_{liq.fossil} & \Delta^{14}C_{coal} \\ \delta^{13}C_{bb} & \delta^{13}C_{liq.fossil} & \delta^{13}C_{coal} \\ 1 & 1 & 1 \end{pmatrix} \cdot \begin{pmatrix} f_{bb} \\ f_{liq.fossil} \\ f_{coal} \end{pmatrix} \quad (4)$$

Where $f$ represents the fraction of a given source, and subscripts denote investigated sample, where 'bb' is biomass burning, 'liq.fossil' is liquid fossil and 'coal' is fossil coal. The last row of the equation ensures fulfilment of the mass-balance criterion. The MCMC technique takes into account the variability of the carbon isotopes for pure sources (endmembers) where $\delta^{13}C$ introduces the largest uncertainty (Supplementary Table 2). This variability is the major source of uncertainties for carbon isotope-based source apportionment, since the precision of the isotope measurements in general, and for these samples is high: s.d. $<0.5‰$ for $\delta^{13}C$, and $<50‰$ for $\Delta^{14}C$.

**Data availability.** The observational data that support the findings of this study are available on request from the corresponding author (Ö.G.) and will be available in the Bolin Cenre Database (http://bolin.su.se/data/). EI data for GFED is freely available and can be found on the website: http://www.globalfiredata.org/data.html. The data for total emissions of BC for different emission scenarios of ECLIPSE is freely available from IIASA: http://www.iiasa.ac.at/web/home/research/research-Programs/air/Global_emissions.html.

For an ECLIPSE version with emissions split into fossil and biofuel please contact IIASA directly.

**Code availability.** The FLEXPART model is freely available to the scientific community. It can be accessed under https://www.flexpart.eu/.

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

## Acknowledgements

A.A., Ö.G. and P.W. acknowledge financial support from the Swedish Energy Agency (contract number 35450-2), the European Union under the 7th Framework Programme (project acronym: CSI:ENVIRONMENT; contract number PITN-GA-2010-264329), the Nordic Council of Ministries Defrost project as part of the Nordic Centres of Excellence (NCoE), and the Swedish funding agency Formas (contract number 942-2015-1070). We gratefully acknowledge Tyler Logan and staff of the Swedish Polar Research Secretariat, at the Abisko Research Station (ANS), for assistance in the field. We thank Zbigniew Klimont and Chris Heyes at the International Institute for Applied System Analysis (IIASA) for providing BC emissions from their GAINS model, ECMWF is acknowledged for meteorological data and part of this study (NILU) were funded under the FP7 project ECLIPSE (contract number 282688) and the Norwegian Research Council project SLICFONIA (contract number 233642). We are also grateful to three anonymous reviewers for helpful comments on an earlier version of the manuscript.

## Author contributions

P.W., A.A and Ö.G. designed the observation-based source apportionment and together with A.S. and S.E., the concept of observation-model comparison. Samples were collected by staff of the Swedish Polar Research Secretariat and analysed by P.W. A.A. was in charge of MCMC simulation procedure while S.E. in charge of model simulations. P.W. wrote the paper with input from all co-authors, and produced the figures.

## Additional information

**Competing financial interests:** The authors declare no competing financial interests.

