## [Peer Review File · Nature Communications]

Reviewers' comments:

Reviewer #1 (Remarks to the Author):

The main limitation of this study is the only measurement on the ground based stations, it has been vastly concluded from previous studies (including loads of long term studies) that the lower altitude sources in the Arctic are mainly influenced by high latitude sources, such as Europe; but at higher altitude, the source could originate remotely from lower latitude, such as Asia. Therefore I couldn't see anything novel from this perspective. The long-term isotope carbon measurement may be an interesting part, but the details of data interpretation (how to discriminate from different potential sources) and uncertainty associated with this technology is not fully discussed, leaving the unknown quality of the dataset. There are massive duplications in the supplement, which reads a bit tedious; I would suggest one or two plots merging the similar plots in one plot will work better. Again, the modelling results are not surprising, as we know it will be from high latitude sources; how the model has been improved from only a few data points (you need to show a good statistics)? At last, I don't think this work can fit in the title "The sources of atmospheric black carbon in the European Arctic" by only measuring the BC close to the ground, this grand title should do with the BC at different altitudes and on the snow as well.

Reviewer #2 (Remarks to the Author):

The authors produce bottom-up boreal black carbon(BC) emissions using the FLEXPART model and report significantly better agreement with top-down observations after including the BC contribution from open vegetation fires. This is an interesting and significant result as it helps remedy the inability of contemporary transport models to accurately reproduce Arctic BC loadings and seasonality.

As a heavy user of MODIS fire data to support fire emissions modeling, I have focused my attention on this aspect of the manuscript. In this regard, I believe the authors have overlooked several important details (enumerated below) with respect to their use of MODIS fire counts as described in the section "Estimation of open fire influence using MODIS" (lines 270 - 277).

1) As a very minor point, the authors should state which MODIS Collection they are using. Almost certainly these were Collection 5 data, but for clarity this should be stated explicitly.

2) Following Wotawa et al. (2006), the authors "...assumed that every detection represents a burned area of 180 ha, based on a statistical analysis of MODIS fire detections with independent data on area burned." Wotawa et al. reported four different fire-count-to-burned-area ratios, ranging from 160 to 193 ha per fire count, but these ratios were for Terra MODIS fire pixels only. In the present study the authors include both Terra and Aqua MODIS fire pixels, thus one would expect that the figure of 180 ha adopted by the authors should in fact be ~2x smaller. Please correct and/or justify. Furthermore, Wotawa et al. (2006) used the Collection 4 MODIS product, while the authors presumably (see #1 above) used Collection 5. It is not clear that the Wotawa et al. figures are applicable to Collection 5. Again, please correct and/or justify. In combination these discrepancies

may help account for the 2x overestimation reported by the authors for the fall period.

3) Related to #2 above, the Wotawa et al. fire-count-to-burned-area ratios are applicable to Boreal Asia and Boreal North America. However, many (most?) of the source fires in the present study are agricultural fires located at lower latitudes (~40N), and for such fires a figure of 180 ha (1.8 km²) per fire pixels seems extremely high (agricultural fields are typically ~10x smaller). Please explain and/or justify.

Minor Suggestions

Introduction: It would be helpful to note the 9 September 2011 - 27 March 2013 study period in the main text. Currently this information is buried in Figure 1 but otherwise not revealed until very late in the manuscript (L225).

L33: "are not agreeing"  "does not agree"

L37: Change "times" to "and" since the multiplication is already noted earlier in the sentence ("..the product of...").

L57: "This observational data was"  "These observational data were"

L216: "right"  "proper"

Figure 4: Please replace "10E-7" in axis label with a real superscript.

Reviewer #3 (Remarks to the Author):

The manuscript by Winiger et al. describes a novel approach for apportioning the sources of black carbon aerosols in the Arctic atmosphere using radiocarbon and stable carbon isotope. And the isotope based source apportionment results were fairly comparable with an atmospheric transport model. Their results showed that the model simulation of BC concentrations and source information can be much improved when open biomass burning is incorporated. Although similar approach has been already applied in East/South Asia and the Arctic by the same group, this manuscript presents very interesting and important results and provides insight into the relative importance of biomass burning and fossil combustion emissions to BC in the Arctic atmosphere. The measurements and model predictions are of high quality, the data correction and error analysis is thorough. Therefore, I recommend for a publication if the authors are able to address the revisions and questions I present in the following.

Major comments:

One of the major conclusion is the model could reproduce BC concentrations and source information very well, which was validated by ¹⁴C source apportionment. However, it seems that the model underestimated biomass burning contribution by ~15% compared with ¹⁴C results. If this is a

systematic bias, the author should extend the discussion about possible explanations. For example, are wood burning emissions included in the model? Recent study studies suggest that wood burning for heating is a very important contributor of carbonaceous aerosols in Europe 1-3.

For EC concentrations, the model did not predict very well when EC concentrations were very low (Figure 2, 2011 winter). Why? Is it because of high uncertainty of ^{14}C measurement when EC is low or overestimation of BC emissions in the model? The model (e.g. inventory input) should be described with more details? Without these information, the bias (or difference) between model and observation stills remains unexplained. I think this is the most novelty part of this study.

I noticed that the same author conducted a very similar observational work in a different site in the Arctic 4. So I encourage the authors include the model vs. observation comparisons from those data in the current paper.

Specific comments:

Abstract: detailed source information of BC should be given. For example, how much contribution are from biomass burning or fossil fuel combustion? Lines 17-19: It should be noted that high correlation coefficient does not necessarily mean a good agreement. Slope and offset should be given (at least should be discussed in the main paper).

Lines 52-53: Some references (e.g. 5, 6) are missing here.

Lines 79-81: Secondary organic carbon from fossil fuel emission could increase OC/EC ratios as well.

Lines 117-118. Please give the reason of a larger seasonal variability in summer. Is it partially due to ^{14}C measurement uncertainty as BC concentrations are very low in summer?

Lines 132-138: see my comments above. Is R from a linear regression or others?

Lines 146-148: see my comments above. The model seems to consistently underestimate biomass burning contribution. Why? More details about inventory and ^{14}C measurement uncertainties should be included in the discussion.

Lines 240-248: The measurement of ^{13}C and ^{14}C in EC is derived from the EC part separated from TOT method. The isotope data present here only represent EC which was isolated by the current thermal method. This EC part may not fully represent the nature of BC discussed in the study. Did the authors evaluate ^{14}C measurement uncertainty from possible EC loss (negative artifact) and charring (positive artifact) 7, 8? If not, this should be at least mentioned in the method section.

References

1. Fuller, G. W.; Tremper, A. H.; Baker, T. D.; Yttri, K. E.; Butterfield, D., Contribution of wood burning to PM₁₀ in London. *Atmos. Environ.* 2014, 87, 87-94.
2. Herich, H.; Gianini, M. F. D.; Piot, C.; Mocnik, G.; Jaffrezo, J. L.; Besombes, J. L.; Prevot, A. S. H.; Hueglin, C., Overview of the impact of wood burning emissions on carbonaceous aerosols and PM in large parts of the Alpine region. *Atmos. Environ.* 2014, 89, 64-75.
3. Genberg, J.; Hyder, M.; Stenström, K.; Bergström, R.; Simpson, D.; Fors, E.; Jönsson, J. Å.; Swietlicki, E., Source apportionment of carbonaceous aerosol in southern Sweden. *Atmos. Chem. Phys.* 2011, 11, (22), 11387-11400.
4. Winiger, P.; Andersson, A.; Yttri, K. E.; Tunved, P.; Gustafsson, Ö., Isotope-Based Source Apportionment of EC Aerosol Particles during Winter High-Pollution Events at the Zeppelin Observatory, Svalbard. *Environ. Sci. Technol.* 2015, 49, (19), 11959-11966.

5. Zhang, Y. L.; Schnelle-Kreis, J.; Abbaszade, G.; Zimmermann, R.; Zotter, P.; Shen, R. R.; Schafer, K.; Shao, L.; Prevot, A. S.; Szidat, S., Source Apportionment of Elemental Carbon in Beijing, China: Insights from Radiocarbon and Organic Marker Measurements. *Environ. Sci. Technol.* 2015, 49, (14), 8408-15.
6. Pavuluri, C. M.; Kawamura, K.; Uchida, M.; Kondo, M.; Fu, P. Q., Enhanced modern carbon and biogenic organic tracers in Northeast Asian aerosols during spring/summer. *J. Geophys. Res.* 2013, 118, (5), 2362-2371.
7. Zhang, Y. L.; Perron, N.; Ciobanu, V. G.; Zotter, P.; Minguillón, M. C.; Wacker, L.; Prévôt, A. S. H.; Baltensperger, U.; Szidat, S., On the isolation of OC and EC and the optimal strategy of radiocarbon-based source apportionment of carbonaceous aerosols. *Atmos. Chem. Phys.* 2012, 12, (22), 10841-10856.
8. Mouteva, G. O.; Fahrni, S. M.; Santos, G. M.; Randerson, J. T.; Zhang, Y. L.; Szidat, S.; Czimczik, C. I., Accuracy and precision of ^{14}C -based source apportionment of organic and elemental carbon in aerosols using the Swiss_4S protocol. *Atmos. Meas. Tech.* 2015, 8, (9), 3729-3743.

Author responses to reviews and edits to manuscript NCOMMS-16-05930

" The sources of atmospheric black carbon at a European gateway to the Arctic "

by Winiger, Patrik; Andersson, August; Eckhardt, Sabine; Stohl, Andreas and Gustafsson, Örjan

We thank all three Reviewers and the Editor for their careful reading of our manuscript and overall supportive comments and suggestions. This constructive input has contributed to improve the manuscript significantly, where we have also had a special eye to improve clarity.

First of all, we agree with Reviewer 1 to have a more specific title, which now reads '*The sources of atmospheric black carbon at a European gateway to the Arctic*'. We have also detailed the Methods description for further clarity. Reviewer 2 brought up a concern regarding the fire emissions estimates from satellite products; we have now updated these model calculations using the more well-documented GFED (Global Fire Emissions Database, version 4.1s¹) product – with little changes to the overall results (Figure 1 in the response letter). Reviewer 3 and the Editor asked about the possibility to apply the present modelling approach also to our previously published results from Svalbard (*Winiger et al. 2015*)². While there are several significant differences in scope and sample/time selection between the short-term earlier study (focus/bias on short high-pollution events spread over only several weeks) and the current study (18-mo continuous complete coverage), we have now performed the extensive and labour-intensive bottom-up modelling for the Zeppelin high-pollution events reported in the earlier paper. The results, now included in the revised ms, demonstrate a larger offset between model and observations for the short-term high-pollution events on Zeppelin, than what is observed in the longer-term observations of the current ms. As now discussed in new text, the small data set on short-term (~24h) high-loading episodes likely reflect more local contributions from wintertime biomass burning sources, likely explaining the offset. For non-elevated episodes, the modelled and observed black carbon concentrations are in a good agreement (see more below). Reviewers 1 and 2 also bring up the issue of the uncertainties related to the carbon isotope-based source apportionment, upon which we have now expanded our description. These and all other issues raised by the Reviewers are detailed below, organized such that first the reviewer/editor comments are given in *italic*, directly followed by our detailed response and outline of resulting edit in (blue coloured) regular font. References in our responses to line numbers in the manuscript refer to the line numbers of the new version.

Editor's Comments:

While all concerns must be addressed, please pay particular attention to the reviewer comments regarding the justification/validation of remote sensing techniques, as well as clarification of the methods as a whole (please note text limitations outlined below). We also encourage you to consider providing a comparison to previously published works, as suggested by R#3.

We thank the Editor for specific suggestions to improve the quality and clarity of the manuscript. We reviewed the manuscript in detail to improve the description of our methods. To address Reviewer 2's comment regarding the usage of remote sensing data to estimate BC fire emissions, we have rerun our model calculations using the well-established GFED product (Global Fire Emissions Database, version 4.1s¹), which was not available yet during our first efforts. Thanks to a comment by Reviewer 2 we realized a small error in the treatment of BC emission data, namely the double accounting of agricultural waste burning, which was included in both, the fossil fraction, as well as (at least partially) in the fire emission data received by satellite. We have now corrected this error.

In response letter Figure 1 (below) we show three different scenarios from the model calculations: 1. Original model, 2. Original model with corrected agricultural waste burning and 3. New model with GFED4.1s fire emissions and corrected agricultural waste burning. It is clear that these corrections did not affect the results in any major way, and the conclusions remain unchanged. In addition, we performed the

comparison with previously published results (*Winiger et al., 2015*)². Due to data selection bias for that limited study (capturing only events with elevated BC concentrations), which did not offer any comparison with models, it is hard to evaluate the mismatch between the now performed modelling and observations for the former case. We also emphasize that the Winiger et al. study was conducted over a limited time-period (January to March 2009), whereas the present study investigated the full year cycle (18-mo continuous).

We welcome the Editor's invitation to include more material in the main manuscript and have included Figure 5 (previously in the Supplementary Information) as well as extended the methods sections on carbon isotopes and Bayesian modelling, in addition to changes made due to all reviewer comments.

Response Letter Figure 1. Comparison of three different model runs. a, The original model run, with double counting of agricultural waste burning (AWB). **b,** The original model run with AWB only in the biomass burning fraction. **c,** Final version with GFED replacing the previous fire algorithm and AWB covered by GFED emissions.

Reviewer: 1

The main limitation of this study is the only measurement on the ground based stations, it has been vastly concluded from previous studies (including loads of long term studies) that the lower altitude sources in the Arctic are mainly influenced by high latitude sources, such as Europe; but at higher altitude, the source could originate remotely from lower latitude, such as Asia. Therefore, I couldn't see anything novel from this prospective. The long-term isotope carbon measurement may be an interesting part, but the details of data interpretation (how to discriminate from different potential sources) and uncertainty associated with this technology is not fully discussed, leaving the unknown quality of the dataset. There are massive duplications in the supplement, which reads a bit tedious; I would suggest one or two plots merging the similar plots in one plot will work better. Again, the modelling results are not surprising, as we know it will be from high latitude sources; how the model has been improved from only a few data points (you need to show a good statistics)? At last, I don't think this work can fit in the title "The sources of atmospheric black carbon in the European Arctic" by only measuring the BC close to the ground, this grand title should do with the BC at different altitudes and on the snow as well.

We thank Reviewer 1 for the assessment of our manuscript and for providing suggestions to improve clarity and quality. We agree in a general sense with the reviewer that it would be valuable to have the year-round continuous Arctic BC and 14C-based BC source apportionment data also in a vertical columnar resolution, yet this is unfortunately not achievable in the present (for anywhere on Earth). Meanwhile, research at ground-based observatories around the Arctic is continuing to provide useful new knowledge of aerosols in the Arctic. A long-standing challenge in Arctic atmospheric science is the inability of models to capture the seasonal cycle of BC and other atmospheric constituents at ground-based stations despite multiple efforts in recent years³⁻⁷. Thus, even though the emission sources contributing to ground-level concentrations qualitatively may be more regional than at higher altitudes, our quantitative understanding of how these sources contribute to the Arctic have been poor. The current study, for the first time, show good quantitative agreement between observations and modelling at an Arctic site, both for concentrations and for estimated sources. Central to this agreement is the inclusion of fire emission estimates from remote sensing products in the modelling. This shows that the FLEXPART model well captures the transport of BC, but that a correct assessment of emissions is key to obtain closure between models and observations. These findings suggest increased confidence for future chemical transport and climate modelling efforts. We have paid special attention to clarify this message in the manuscript (lines 133-137;166-169 & 208-211).

We thank Reviewer 1 for highlighting lack of clarity regarding the description of the carbon isotope techniques and related uncertainties. The related uncertainties: experimental and source estimation uncertainties were estimated using Monte Carlo simulations, and are typically low (<5%) for estimated fraction biomass. For the differentiation between liquid fossil fuels and fossil coal the uncertainties are larger due to the larger uncertainties related to what is known about the stable carbon ($\delta^{13}\text{C}$) signatures of the pure source emissions (endmembers). We have re-written the methods section to increase clarity (lines 278-280 & 333-343).

We agree that the supplementary materials were long and repetitive. To improve clarity and distinctness we have moved Supplementary Figure S 39 to the main manuscript (Figure 5), and improved the quality and reduced the number of the remaining supplementary figures (see updated Supplementary Information).

The linear fit between model and observations, both for concentrations and relative source contributions ($R^2=0.61$ and $R^2=0.57$, respectively), is significantly improved when the fire emissions from the remote sensing product is included with the bottom-up emission inventory (coefficient of determination (R^2) of source contribution without open fire emissions = 0.26). We have emphasized this feature in the main manuscript (lines 166-169).

Finally, we agree with the Reviewer that the title maybe too generalized. We have changed it to: '*The sources of atmospheric black carbon at a gateway to the European Arctic*'.

Reviewer: 2

The authors produce bottom-up boreal black carbon(BC) emissions using the FLEXPART model and report significantly better agreement with top-down observations after including the BC contribution from open vegetation fires. This is an interesting and significant result as it helps remedy the inability of contemporary transport models to accurately reproduce Arctic BC loadings and seasonality.

We appreciate that Reviewer 2 recognizes the significance of our findings and thank the Reviewer for insightful comments and interest in the presented work.

As a heavy user of MODIS fire data to support fire emissions modelling, I have focused my attention on this aspect of the manuscript. In this regard, I believe the authors have overlooked several important details (enumerated below) with respect to their use of MODIS fire counts as described in the section "Estimation of open fire influence using MODIS" (lines 270 - 277).

Reviewer 2 has raised an important issue here. The application of the simplistic algorithm used in this work was undertaken because other fire products (i.e. Global Fire Emissions Database, GFED4) were not available at that time. Since GFED4 has now become available, which is - as the Reviewer most likely knows - a widely accepted fire emissions inventory¹. We have now gone back and replaced the previously used MODIS data with GFED, version GFED4.1s. This has of course led to changes in the methods description of manuscript (Figures, Supplementary Figures and Tables). However, we would like to stress, that it did not lead to any major changes in the results or conclusions of the manuscript, as can be seen in Response Letter Response Letter Figure 1 (above).

We replace the inset "a" in Figure 1 with fire BC emission data from GFED4.1s and the units of the main map have been changed to facilitate direct comparison for the reader. We updated Figures 2 & 3, as well as Supplementary Figs. 3 & 21 and Supplementary Tables 4 & 8. Finally, the method section regarding the contribution of satellite-derived fire BC has been completely revised (lines 324-331).

1) As a very minor point, the authors should state which MODIS Collection they are using. Almost certainly these were Collection 5 data, but for clarity this should be stated explicitly.

This is correct; we did use Collection 5 data. As you probably know, with GFED4.1s, we now implement Collection 5.1 data, as stated in the newly revised method section (lines 324-331).

2) Following Wotawa et al. (2006), the authors "...assumed that every detection represents a burned area of 180 ha, based on a statistical analysis of MODIS fire detections with independent data on area burned." Wotawa et al. reported four different fire-count-to-burned-area ratios, ranging from 160 to 193 ha per fire count, but these ratios were for Terra MODIS fire pixels only. In the present study the authors include both Terra and Aqua MODIS fire pixels, thus one would expect that the figure of 180 ha adopted by the authors should in fact be ~2x smaller. Please correct and/or justify. Furthermore, Wotawa et al. (2006) used the Collection 4 MODIS product, while the authors presumably (see #1 above) used Collection 5. It is not clear that the Wotawa et al. figures are applicable to Collection 5. Again, please correct and/or justify. In combination these discrepancies may help account for the 2x overestimation reported by the authors for the fall period.

We regret having caused confusion by not properly discussing this issue in our methods section. With the use of GFED4.1s data, this should now be resolved as GFED is more well-documented^{1,15,16}. It did however not solve the conundrum of the overestimated fall periods.

3) Related to #2 above, the Wotawa et al. fire-count-to-burned-area ratios are applicable to Boreal Asia and Boreal North America. However, many (most?) of the source fires in the present study are agricultural fires located at lower latitudes (~40N), and for such fires a figure of

180 ha (1.8 km²) per fire pixels seems extremely high (agricultural fields are typically ~10x smaller). Please explain and/or justify.

Again, with the application of GFED4.1s data, this should now be resolved, since it specifically includes also small fires, as opposed to earlier GFED versions, e.g., GFED3, which used a different filtering scheme.

Minor Suggestions

Introduction: It would be helpful to note the 9 September 2011 - 27 March 2013 study period in the main text. Currently this information is buried in Figure 1 but otherwise not revealed until very late in the manuscript (L225).

Thank you, this has been added, see lines 73-74: "The study took place from 9th September 2011 to 27th March 2013, where the..."

L33: "are not agreeing"  "does not agree"

This sentence was changed to the following (lines 32-34): "... the relative contributions of biomass burning vs. fossil fuel combustion predicted by EI models do not agree with ¹⁴C-based diagnostic source apportionment of BC ..."

L37: Change "times" to "and" since the multiplication is already noted earlier in the sentence ("..the product of...").

Changed (line 37).

L57: "This observational data was"  "These observational data were"

Changed (line 59).

L216: "right"  "proper"

Changed (line 231).

Figure 4: Please replace "10E-7" in axis label with a real superscript.

Changed

Reviewer: 3

The manuscript by Winiger et al. describes a novel approach for apportioning the sources of black carbon aerosols in the Arctic atmosphere using radiocarbon and stable carbon isotope. And the isotope based source apportionment results were fairly comparable with an atmospheric transport model. Their results showed that the model simulation of BC concentrations and source information can be much improved when open biomass burning is incorporated. Although similar approach has been already applied in East/South Asia and the Arctic by the same group, this manuscript presents very interesting and important results and provides insight into the relative importance of biomass burning and fossil combustion emissions to BC in the Arctic atmosphere. The measurements and model predictions are of high quality, the data correction and error analysis is thorough. Therefore, I recommend for a publication if the authors are able to address the revisions and questions I present in the following.

We thank Reviewer 3 for the overall positive assessment. The reviewer's recognition that we here apply the same radiocarbon source apportionment methodology to the Arctic as we have earlier done to BC aerosols in India and China is correct. However, the resulting quantitative source apportionment for EC is very different between India vs China vs the Arctic. We are therefore glad to see that Reviewer 3

acknowledges that "*this manuscript presents very interesting and important results*". Responses to the specific comments/questions are detailed below.

Major comments:

One of the major conclusion is the model could reproduce BC concentrations and source information very well, which was validated by ^{14}C source apportionment. However, it seems that the model underestimated biomass burning contribution by ~15% compared with ^{14}C results. If this is a systematic bias, the author should extent the discussion about possible explanations. For example, are wood burning emissions included in the model? Recent study studies suggest that wood burning for heating is a very important contributor of carbonaceous aerosols in Europe 1-3.

Looking at the total average, the model did underestimate biomass burning by some degree (13% for the reviewed manuscript and 12% in case of the updated GFED4 version of the manuscript). Considering single samples, the model estimates varied from 55-136% to observed fractions of biomass burning. A perhaps more likely explanation, than a systematic bias, is therefore perhaps the inherent uncertainties of the modelling: 12-13% is a rather low offset considering the typically larger uncertainties reported in bottom-up emission inventories¹⁷. A direct observation to model comparison of black carbon concentration, as well as fraction biomass burning, has now been included in Supplementary Table 8.

Wood burning emissions, as our study (and our previous work²) shows, are indeed important contributors to carbonaceous aerosols. Its emissions are included in the model as biofuels, as well as within the GFED4 data, as open biomass burning emissions, including agricultural waste burning.

For more clarity, wood burning is specifically mentioned as biofuel in the description of the model in line 315. Furthermore, we added, detailed discussion in the methods section of the revised manuscript (lines 319-322) and added a detailed table of the ECLIPSE emissions partitioning in Supplementary Table 9.

For EC concentrations, the model did not predict very well when EC concentrations were very low (Figure2, 2011 winter). Why? Is it because high uncertainty of ^{14}C measurement when EC is low or overestimation of BC emissions in the model? The model (e.g. inventory input) should be described with more details? Without these information, the bias (or difference) between model and observation stills remains unexplained. I think this is the most novelty part of this study.

The discrepancy between model and observations during the winter of 2011 is of course important to consider. We currently do not have any clear explanation for the offset in concentrations, yet it should be stressed that the offset in estimated fraction biomass is comparably low. We emphasize that the observed difference in winter 2011 concentrations is not very large in the given context – agreement between models and observations is in general a difficult problem. The uncertainties in ^{14}C measurements are low for this data set (<5%); the amount of matter collected on the filters rather than atmospheric concentrations are what governs this feature – this is why we used long collection times (2-3 weeks).

We have included an expanded description of the bottom-up emission inventory (ECLIPSE) used for this study (lines 301-322). In addition, we have expanded the discussion regarding the offsets in BC concentrations (while similar in fraction biomass) in the manuscript (lines 140-143).

I noticed that the same author conducted a very similar observational work in a different site in the Arctic 4. So I encourage the authors include the model vs. observation comparisons from those data in the current paper.

Yes, the $\Delta^{14}\text{C}$ signature of BC for our earlier study at the Zeppelin Observatory in Svalbard for the period January to March 2009² is compared in the submitted ms. Despite the application of the same ^{14}C technique, there are substantial differences between the observational aspects of these two studies. For the earlier study, we did not have the long filter collections as in Abisko, but only a small set of 24-hours samples. The objective of that study was to understand source contributions to specific high pollution

events. To obtain enough carbon material for the technical radioisotope analysis of these samples we therefore selected samples with high EC concentrations only. On average, those samples displayed a rather high fraction biomass ($52 \pm 15\%$) for a winter period. Despite the differences in scopes and time duration (few 24.h collections vs. 18-months continuous coverage), we have followed the reviewer request, and now used the current FLEXPART-ECLIPSE-GFED model to compare the Svalbard observations with these new modelling results (Response Letter Figure 2). Overall, the model suggested lower concentrations than obtained in those short-term high-pollution observations. We believe that a major reason for this offset is that the time points investigated by carbon-14 were influenced by biomass burning events, which are difficult to capture by the model. The ‘non-events’ in the earlier observation record were in similar range as the model predictions in terms of concentrations. We emphasize that high loading events are not likely to be large fires – which should be incorporated by GFED4 – since these are very scarce in the winter period. Furthermore, we again emphasize that the Svalbard study covers only a limited time-period and is biased by high pollution events – in contrast to the current continuous year-round study in Abisko.

We have now included short comments regarding the Svalbard comparison in the main manuscript (lines 220-227), and an elaboration with a figure in the SI (Supplementary Figure 7 and Supplementary Note 1, lines 195-216 in SI).

Figure 2 Model - observation comparison for Zeppelin a, Elemental (black) carbon concentrations, measured at the Zeppelin Observatory. Original dataset by Yttri *et al.* (2014)¹⁸ (cyan circles), from which samples with highest EC concentration ('Yttri & Winiger') have been selected for radiocarbon

measurements² (black diamonds). Model estimates by FLEXPART (red circles). **b**, Estimated BC source contribution of all FLEXPART samples, measured at the Zeppelin Observatory (blue star).

Specific comments:

Abstract: detailed source information of BC should be given. For example, how much contribution are from biomass burning or fossil fuel combustion? Lines 17-19: It should be noted that high correlation coefficient does not necessarily mean a good agreement. Slope and offset should be given (at least should be discussed in the main paper).

The length of the abstract in *Nature Communications* is limited to 150 words. It is therefore difficult include all relevant information. However, we agree with the Reviewer and have now commented on the quality of the fit, noting that the observed low p-value adds confidence to our interpretation (line 18).

Lines 52-53: Some references (e.g. 5, 6) are missing here.

The two lines in question are on the topic of radiocarbon measurements in BC (EC). The study of reference 5 is very similar to our reference 20 (of the same first author/group as in ref. 5). Reference 6 on the other hand is not on the topic of radiocarbon in black carbon, but in total carbon (TOC) and water soluble organic carbon (WSOC). We do hence not see any reason to include these references here.

Lines 79-81: Secondary organic carbon from fossil fuel emission could increase OC/EC ratios as well.

We agree. The following sentence has been added (lines 79-81):

"OC may come from both primary and secondary sources, and these, in turn, can be from both combustion and non-combustion sources. High OC/EC ratios are therefore..."

Lines 117-118. Please give the reason of a larger seasonal variability in summer. Is it partially due to 14C measurement uncertainty as BC concentrations are very low in summer?

We apologize for this misunderstanding. The variability of the fraction of biomass burning (based on radiocarbon measurements) is not bigger in summer than in winter, nor does the uncertainty of this measurement change much. What this sentence was supposed to mean is, that there are large changes of fraction biomass burning throughout the year. This Sentence has been changed now (line 122-123):

"...large seasonal variability in the fraction of biomass burning contribution was observed throughout the year, ..."

The uncertainty of the radiocarbon method is now discussed in more detail (lines 278-280).

Lines 132-138: see my comments above. Is R from a liner regression or others?

We used linear regressions at all times. This information is now included in the manuscript (line 88-89):

".... all coefficients of determination (R^2) used in this work are from linear regressions. "

Lines 146-148: see my comments above. The model seems to consistently underestimate biomass burning contribution. Why? More details about inventory and 14C measurement uncertainties should be included in the discussion.

As discussed above, the model does not consistently underestimate biomass burning. We agree however that more detail is needed about the inventory and extended the methods section explaining the model set up (lines 316-331).

Lines 240-248: The measurement of 13C and 14C in EC is derived from the EC part separated from TOT method. The isotope data present here only represent EC which was isolated by the current thermal method. This EC part may not fully represent the nature of BC discussed in the study. Did the authors evaluate 14C measurement uncertainty from possible EC loss (negative

artifact) and charring (positive artifact)^{7, 8}? If not, this should be at least mentioned in the method section.

A common aspect, that is well recognized in the literature (including discussed in several of our earlier papers), is the possibility that method-intrinsic pyrogenic carbon is exchanging with ambient elemental carbon in the elution. We have - in *Winiger et al. (2015)*² - performed a detailed sensitivity analysis for the feature, tailored specifically for this type of Arctic studies in our earlier publication². We agree that this should be commented on also in the current manuscript.

It should be noted that the herein employed approach has been thoroughly tested and reported in earlier methods papers and in quite a few published applications papers (incl. *Gustafsson et al., 2009 Science*¹⁹; *Chen et al., 2013, ES&T*²⁰ and *Andersson et al., 2015 ES&T*²¹), where we also acknowledge the involved uncertainties.

The following sentence has thus been added to the methods section (lines 249-252):

" Potential effects of charring with this method, where parts of the OC could end up as pyrogenic carbon in the EC fraction, have been evaluated in earlier work by sensitivity analysis reaching the conclusion that fraction biomass burning could even in extreme cases be overestimated only by a maximum of 7%². "

References

1. Fuller, G. W.; Tremper, A. H.; Baker, T. D.; Yttri, K. E.; Butterfield, D., Contribution of wood burning to PM₁₀ in London. *Atmos. Environ.* 2014, 87, 87-94.
2. Herich, H.; Gianini, M. F. D.; Piot, C.; Mocnik, G.; Jaffrezo, J. L.; Besombes, J. L.; Prevot, A. S. H.; Hueglin, C., Overview of the impact of wood burning emissions on carbonaceous aerosols and PM in large parts of the Alpine region. *Atmos. Environ.* 2014, 89, 64-75.
3. Genberg, J.; Hyder, M.; Stenström, K.; Bergström, R.; Simpson, D.; Fors, E.; Jönsson, J. Å.; Swietlicki, E., Source apportionment of carbonaceous aerosol in southern Sweden. *Atmos. Chem. Phys.* 2011, 11, (22), 11387-11400.
4. Winiger, P.; Andersson, A.; Yttri, K. E.; Tunved, P.; Gustafsson, Ö., Isotope-Based Source Apportionment of EC Aerosol Particles during Winter High-Pollution Events at the Zeppelin Observatory, Svalbard. *Environ. Sci. Technol.* 2015, 49, (19), 11959-11966.
5. Zhang, Y. L.; Schnelle-Kreis, J.; Abbaszade, G.; Zimmermann, R.; Zotter, P.; Shen, R. R.; Schafer, K.; Shao, L.; Prevot, A. S.; Szidat, S., Source Apportionment of Elemental Carbon in Beijing, China: Insights from Radiocarbon and Organic Marker Measurements. *Environ. Sci. Technol.* 2015, 49, (14), 8408-15.
6. Pavuluri, C. M.; Kawamura, K.; Uchida, M.; Kondo, M.; Fu, P. Q., Enhanced modern carbon and biogenic organic tracers in Northeast Asian aerosols during spring/summer. *J. Geophys. Res.* 2013, 118, (5), 2362-2371.
7. Zhang, Y. L.; Perron, N.; Ciobanu, V. G.; Zotter, P.; Minguillón, M. C.; Wacker, L.; Prévôt, A. S. H.; Baltensperger, U.; Szidat, S., On the isolation of OC and EC and the optimal strategy of radiocarbon-based source apportionment of carbonaceous aerosols. *Atmos. Chem. Phys.* 2012, 12, (22), 10841-10856.
8. Mouteva, G. O.; Fahrni, S. M.; Santos, G. M.; Randerson, J. T.; Zhang, Y. L.; Szidat, S.; Czimczik, C. I., Accuracy and precision of ¹⁴C-based source apportionment of organic and elemental carbon in aerosols using the Swiss_4S protocol. *Atmos. Meas. Tech.* 2015, 8, (9), 3729-3743.

References:

- (1) Giglio, L.; Randerson, J. T.; van der Werf, G. R. Analysis of daily, monthly, and annual burned area using the fourth-generation global fire emissions database (GFED4). *J. Geophys. Res. Biogeosciences* **2013**, *118* (1), 317–328 DOI: 10.1002/jgrg.20042.
- (2) Winiger, P.; Andersson, A.; Yttri, K. E.; Tunved, P.; Gustafsson, Ö. Isotope-Based Source Apportionment of EC Aerosol Particles during Winter High-Pollution Events at the Zeppelin Observatory, Svalbard. *Environ. Sci. Technol.* **2015**, *49* (19), 11959–11966 DOI: 10.1021/acs.est.5b02644.
- (3) Eckhardt, S.; Quennehen, B.; Olivie, D. J. L.; Berntsen, T. K.; Cherian, R.; Christensen, J. H.; Collins, W.; Crepinsek, S.; Daskalakis, N.; Flanner, M.; et al. Current model capabilities for simulating black carbon and sulfate concentrations in the Arctic atmosphere: a multi-model evaluation using a comprehensive measurement data set. *Atmos. Chem. Phys.* **2015**, *15* (16), 9413–9433 DOI: 10.5194/acp-15-9413-2015.
- (4) Garrett, T. J.; Brattström, S.; Sharma, S.; Worthy, D. E. J.; Novelli, P. The role of scavenging in the seasonal transport of black carbon and sulfate to the Arctic. *Geophys. Res. Lett.* **2011**, *38* (16) DOI: 10.1029/2011GL048221.
- (5) Calvo, A. I.; Alves, C.; Castro, A.; Pont, V.; Vicente, A. M.; Fraile, R. Research on aerosol sources and chemical composition: Past, current and emerging issues. *Atmos. Res.* **2013**, *120–121*, 1–28 DOI: 10.1016/j.atmosres.2012.09.021.
- (6) Arctic Monitoring and Assessment Programme (AMAP). *AMAP Assessment 2015: Black carbon and ozone as Arctic climate forcers*; Oslo, Norway, 2015.
- (7) Fuzzi, S.; Baltensperger, U.; Carslaw, K.; Decesari, S.; Gon, H. D. Van Der; Facchini, M. C.; Fowler, D.; Koren, I.; Langford, B.; Lohmann, U.; et al. Particulate matter, air quality and climate: lessons learned and future needs. *Atmos. Chem. Phys.* **2015**, *15*, 8217–8299 DOI: 10.5194/acp-15-8217-2015.
- (8) Jacob, D. J.; Crawford, J. H.; Maring, H.; Clarke, A. D.; Dibb, J. E.; Emmons, L. K.; Ferrare, R. A.; Hostetler, C. A.; Russell, P. B.; Singh, H. B.; et al. The Arctic Research of the Composition of the Troposphere from Aircraft and Satellites (ARCTAS) mission: design, execution, and first results. *Atmos. Chem. Phys.* **2010**, *10* (11), 5191–5212 DOI: 10.5194/acp-10-5191-2010.
- (9) Brock, C. A.; Cozic, J.; Bahreini, R.; Froyd, K. D.; Middlebrook, A. M.; McComiskey, A.; Brioude, J.; Cooper, O. R.; Stohl, A.; Aikin, K. C.; et al. Characteristics, sources, and transport of aerosols measured in spring 2008 during the aerosol, radiation, and cloud processes affecting Arctic Climate (ARCPAC) Project. *Atmos. Chem. Phys.* **2011**, *11* (6), 2423–2453 DOI: 10.5194/acp-11-2423-2011.
- (10) Stone, R. S.; Herber, A.; Vitale, V.; Mazzola, M.; Lupi, A.; Schnell, R. C.; Dutton, E. G.; Liu, P. S. K.; Li, S.-M.; Dethloff, K.; et al. A three-dimensional characterization of Arctic aerosols from airborne Sun photometer observations: PAM-ARCMIP, April 2009. *J. Geophys. Res.* **2010**, *115* (D13), D13203 DOI: 10.1029/2009JD013605.
- (11) Schwarz, J. P.; Spackman, J. R.; Gao, R. S.; Watts, L. A.; Stier, P.; Schulz, M.; Davis, S. M.; Wofsy, S. C.; Fahey, D. W. Global-scale black carbon profiles observed in the remote atmosphere and compared to models. *Geophys. Res. Lett.* **2010**, *37* (18), n/a-n/a DOI: 10.1029/2010GL044372.
- (12) Schwarz, J. P.; Samset, B. H.; Perring, A. E.; Spackman, J. R.; Gao, R. S.; Stier, P.; Schulz, M.; Moore, F. L.; Ray, E. A.; Fahey, D. W. Global-scale seasonally resolved black carbon vertical profiles over the Pacific. *Geophys. Res. Lett.* **2013**, *40* (20), 5542–5547 DOI: 10.1002/2013GL057775.
- (13) Wofsy, S. C. HIAPER Pole-to-Pole Observations (HIPPO): fine-grained, global-scale measurements of climatically important atmospheric gases and aerosols. *Philos. Trans. R. Soc. A Math. Phys. Eng. Sci.* **2011**, *369* (1943), 2073–2086 DOI: 10.1098/rsta.2010.0313.
- (14) Warneke, C.; Froyd, K. D.; Brioude, J.; Bahreini, R.; Brock, C. A.; Cozic, J.; de Gouw, J. A.; Fahey, D. W.; Ferrare, R.; Holloway, J. S.; et al. An important contribution to springtime Arctic aerosol from biomass burning in Russia. *Geophys. Res. Lett.* **2010**, *37* (1), n/a-n/a DOI:

- 10.1029/2009GL041816.
- (15) Giglio, L.; Loboda, T.; Roy, D. P.; Quayle, B.; Justice, C. O. An active-fire based burned area mapping algorithm for the MODIS sensor. *Remote Sens. Environ.* **2009**, *113* (2), 408–420 DOI: 10.1016/j.rse.2008.10.006.
 - (16) van der Werf, G. R.; Randerson, J. T.; Giglio, L.; Collatz, G. J.; Mu, M.; Kasibhatla, P. S.; Morton, D. C.; DeFries, R. S.; Jin, Y.; van Leeuwen, T. T. Global fire emissions and the contribution of deforestation, savanna, forest, agricultural, and peat fires (1997–2009). *Atmos. Chem. Phys.* **2010**, *10* (23), 11707–11735 DOI: 10.5194/acp-10-11707-2010.
 - (17) Zhao, Y.; Nielsen, C. P.; Lei, Y.; McElroy, M. B.; Hao, J. Quantifying the uncertainties of a bottom-up emission inventory of anthropogenic atmospheric pollutants in China. *Atmos. Chem. Phys.* **2011**, *11* (5), 2295–2308 DOI: 10.5194/acp-11-2295-2011.
 - (18) Yttri, K. E.; Lund Myhre, C.; Eckhardt, S.; Fiebig, M.; Dye, C.; Hirdman, D.; Ström, J.; Klimont, Z.; Stohl, a. Quantifying black carbon from biomass burning by means of levoglucosan - A one-year time series at the Arctic observatory Zeppelin. *Atmos. Chem. Phys.* **2014**, *14* (12), 6427–6442 DOI: 10.5194/acp-14-6427-2014.
 - (19) Gustafsson, O.; Kruså, M.; Zencak, Z.; Sheesley, R. J.; Granat, L.; Engström, E.; Praveen, P. S.; Rao, P. S. P.; Leck, C.; Rodhe, H. Brown clouds over South Asia: biomass or fossil fuel combustion? *Science* **2009**, *323* (5913), 495–498 DOI: 10.1126/science.1164857.
 - (20) Chen, B.; Andersson, A.; Lee, M.; Kirillova, E. N.; Xiao, Q.; Kruså, M.; Shi, M.; Hu, K.; Lu, Z.; Streets, D. G.; et al. Source forensics of black carbon aerosols from China. *Environ. Sci. Technol.* **2013**, *47* (16), 9102–9108 DOI: 10.1021/es401599r.
 - (21) Andersson, A.; Deng, J.; Du, K.; Yan, C.; Zheng, M.; Sköld, M.; Gustafsson, O. Regionally-varying combustion sources of the January 2013 severe haze events over eastern China. *Environ. Sci. Technol.* **2015**, *49* (4), 2038–2043 DOI: 10.1021/es503855e.

Reviewers' comments:

Reviewer #1 (Remarks to the Author):

I think this paper has been improved and more thoroughly addressed the comparison between model and observation, though it is not conclusive. I would suggest to make the plots clearer how the model has been improved by introducing the updated inventory.

Many plots still need improvement: in Fig. 2, is it possible to provide some more statistics on y-axis, like standard deviation, percentiles. I still feel struggled to really understand Fig.3, there are too many legends and information on a single plot, could we add some value on it or just a few more 1-D plots? What is the colour scale for Fig. 4? There is no need different markers in Fig. 5. Why the standard deviation for fossil fuel and coal are the same? Can both plots be merged into one?

I would suggest the discussion part is more conclusive, such as what are the findings here to improve model activities?

A few very related references should be added and discussed:

doi:10.5194/acp-15-10057-2015

doi:10.5194/acp-15-11537-2015

Reviewer #2 (Remarks to the Author):

I believe the authors have satisfactorily addressed the issues raised during the initial reviews. In particular, the switch to GFED4s burned area has ameliorated my original concerns regarding the use of MODIS fire data. As an aside, I am not entirely comfortable with the way small fires have been included in GFED4s, but this is not an issue of any significance for the present work. I recommend publication as is.

Minor Suggestions

Figure 4 and supplementary Figures 1, 4, 6, and 7b: Add units of degrees to axis labels.

Reviewer #3 (Remarks to the Author):

Publish as is.

Author responses to reviews and resulting edits to manuscript NCOMMS-16-05930A

" The sources of atmospheric black carbon at a European gateway to the Arctic "

by Winiger, Patrik; Andersson, August; Eckhardt, Sabine; Stohl, Andreas and Gustafsson, Örjan

We thank all three Reviewers and the Editor for their repeated reading of our manuscript and for tentatively accepting it for publication, under consideration of further minor changes.

All responses and changes are detailed below, organized such that first the reviewer comments are given in *italic*, directly followed by our detailed response and outline of resulting edit in (blue coloured) regular font.

Reviewer: 1

I think this paper has been improved and more thoroughly addressed the comparison between model and observation, through it is not conclusive. I would suggest to make the plots clearer how the model has been improved by introducing the updated inventory.

Many plots still need improvement: in Fig. 2, is it possible to provide some more statistics on y-axis, like standard deviation, percentiles. I still feel struggled to really understand Fig.3, there are too many legends and information on a single plot, could we add some value on it or just a few more 1-D plots? What is the colour scale for Fig. 4? There is no need different markers in Fig. 5. Why the standard deviation for fossil fuel and coal are the same? Can both plots be merged into one?

I would suggest the discussion part is more conclusive, such as what are the findings here to improve model activities?

A few very related references should be added and discussed:

doi:10.5194/acp-15-10057-2015

doi:10.5194/acp-15-11537-2015

We thank Reviewer 1 for reconsideration of our manuscript. We are glad that this reviewer thinks the manuscript has been improved.

Regarding the question of adding standard deviation of the model predictions to Fig. 2, this has previously been considered but discouraged, since there is currently no way of addressing standard deviations in models like this. One of several reasons is that the underlying emission inventory is only available with single central values – with no indications of the uncertainty of this important component.

However, we have now added the standard deviation of the elemental carbon (EC) concentration of the composites/samples (observation) to the revised Fig. 2. This information was previously only shown in Supplementary Table 9.

Fig.3 is similar to Fig. 2 in the previous and present version, with the added new information being the $\delta^{13}\text{C}$ -fingerprint of each sample in panel **a** and the model-based source fractions of fossil fuel, biofuel and fire (biomass) in panel **b**. The complexity of the figure can help the reader to identify single samples on the two-dimensional isotope plot, which otherwise may be a bit more difficult.

The colour scale in Fig. 4 is the logarithm to the base of 2 (\log_2). This information has been added to the figure caption of Fig. 4.

The different markers in Fig. 5 panel **b** make it easier for the reader to distinguish coal and liq.fossil which have similar standard deviations because "... these two have almost equal uncertainties in their isotopic signature (Supplementary Table 2)", as it says in the manuscript (line 198). Merging these

plots into one would unfortunately create a lot of overlapping areas, making it harder to distinguish the data.

This reviewer suggests that the Discussions part may be more “conclusive”. We have again revisited the relevant paragraphs and made only minor edits. It seems to us that the text, without being repetitive of earlier parts of the manuscript, is clearly communicating the main findings of the manuscript, especially w.r.t. the observation-model comparison. The final part states: “...*systematic biases and uncertainties in BC emission inventories, possibly enhanced by challenges in modelling of the transport in the Arctic troposphere and in the scavenging of aerosols, have caused the earlier model-observation mismatch of BC in the Arctic. The present isotope-constrained source apportionment study now demonstrates the ability of the FLEXPART model, with improved description of BC emissions, to reproduce both the absolute concentrations and their seasonal amplitude, as well as assigning the contribution of different source classes to the simulated BC in agreement with the observed source diagnostics.*” (lines 222-228).

Finally, we thank the reviewer for pointing out these two additional studies. Raatkinen et al. 2015 was previously considered, but then not referenced because the authors used rBC (refractive black carbon), which would have just introduced another (unrelated) form of BC measurement. However, that study does offer some insight with regard to source regions and has thus now been added and discussed (page 9, line 182). In general, the use of high altitude aircraft BC studies, such as in the proposed Liu et al. 2015 (also rBC), is limited because they do not offer a "complete" BC record but only a short snapshot in time and could potentially be biased towards sampling of (high BC concentration) plumes. This does of course not mean, that this was the case in Liu et al. 2015. We have however followed the reviewer recommendation and included also this reference at an appropriate place (page 9, line 192).

Reviewer: 2

I believe the authors have satisfactorily addressed the issues raised during the initial reviews. In particular, the switch to GFED4s burned area has ameliorated my original concerns regarding the use of MODIS fire data. As an aside, I am not entirely comfortable with the way small fires have been included in GFED4s, but this is not an issue of any significance for the present work. I recommend publication as is.

Minor Suggestions

Figure 4 and supplementary Figures 1, 4, 6, and 7b: Add units of degrees to axis labels.

We thank Reviewer 2 for the second review of our manuscript, especially with regard to this reviewer’s deeper insight into satellite-based fire products.

The minor suggestions on adding units of degrees to figure labels have been implemented in the final version.

Reviewer: 3

Publish as is.

We thank Reviewer 3 for reviewing our manuscript again and recommending it for publication.